# Effect of a Monomer Composition on the Mechanical Properties and Glass Transition Temperature of a Waterborne Polyurethane/Graphene Oxide and Waterborne Polyurethane/MWCNT Nanocomposite

**DOI:** 10.3390/polym12092013

**Published:** 2020-09-03

**Authors:** Hyung Joong Kim, Jihye Han, Younggon Son

**Affiliations:** Division of Advanced Materials Science and Engineering, Kongju National University, Cheonan, Chungnam 31080, Korea; hyungjk@kongju.ac.kr (H.J.K.); gwg2113@naver.com (J.H.)

**Keywords:** waterborne polyurethane, nanocomposite, graphene oxide, cationic surfactant, ionic interaction

## Abstract

Anionic waterborne polyurethane (aWPU) is not compatible with graphene oxide (GO) due to the repulsive force acting on identical ionic charges. In this study, we fabricated cationic surfactant treated GO and cationic surfactant treated carbon nanotube (CNT) to increase the compatibility with aWPU. Cationic waterborne polyurethane (WPU) and nanocomposites thereof were also prepared. On the basis of the mechanical properties of the nanocomposites, glass transition temperature (*T*_g_), and a stability test, it was found that the compatibility between WPU and a nanofiller (NF) was enhanced to a great extent when WPU and NF had opposite ionicity. The *T*_g_ and mechanical properties of WPU increased with the addition of NF, showed the maximum value and thereafter decreased with further addition. The effect of composition of ionic monomer in WPU was also investigated. As the composition of the ionic monomer increases, the concentration of NF for the maximum *T*_g_ and mechanical properties increases. This was attributed to the ionic association between the NF and WPU.

## 1. Introduction

Polyurethane (PU) has various applications including fibers, coatings, adhesives, and smart actuators [1,2,3]. It can be prepared with tailor-made properties by using well-designed combinations of the monomeric materials [4]. For instance, PU/organic solvent mixtures have been widely used as coatings, adhesives, and ink. However, due to growing environmental restrictions on volatile organic compounds, waterborne polyurethane (WPU) has received increasing attention in recent coating and adhesive industries. WPU is a binary colloidal system where PU is microdispersed in an aqueous medium. As a binder, WPU can be a good alternative to the organic solvent-based PU binder.

In spite of these advantages, expansion of the WPU application could be limited due to serious shortcomings, low stiffness, in some fields [5]. These shortcomings can be reduced by the incorporation of rigid, electric conductive, and impermeable nanofillers such as graphene and carbon nanotubes. Accordingly, considerable effort has been made to improve the mechanical, electrical, and barrier properties of WPU by incorporating carbon nanotubes, graphene, functionalized graphene, and graphene oxide (GO).

In order to increase the physical properties of polymer nanocomposites, the homogeneously dispersed structure of nanofillers (NF) is of great importance. Since hydrophobic graphene does not show homogeneous dispersion in hydrophilic WPU, many studies have adopted functionalized graphene or an emulsifier to improve the dispersion of graphene [6]. Many studies have incorporated functional groups on graphene in order to increase its hydrophilicity and investigated its dispersion ability in WPU. However, the results were not satisfactory [7,8,9,10,11,12]. On the other hand, GO itself contains many hydrophilic groups and is water-soluble. Therefore, dispersion of GO in WPU is easier than dispersing rGO (reduced graphene oxide) and functionalized rGO.

Many studies in the literature have used rGO as a nanofiller to improve the electrical, mechanical, and barrier properties of WPU. rGO is still more expensive than multiwalled carbon nanotubes (MWCNTs), and dispersion of rGO in WPU is not easy [13,14,15,16,17,18]. Attempts to increase the electrical conductivity of WPU with GO or rGO have not been successful, and the improvement is inferior to that obtained with MWCNTs [13,14]. Thus, enhancement of electrical conductivity of WPU by rGO is thus not an efficient approach. For some applications, the electrical conductivity is not critical, but the mechanical properties and barrier property are of greater concern. WPU/GO is a good alternative for those applications.

There have been a few studies on WPU/GO nanocomposites. Yoon et al. investigated a WPU/isocyanate modified GO nanocomposite [15]. They incorporated allyl isocyanate onto GO, which subsequently reacts with acrylate termini of WPU. In this way, GO was chemically bonded to polymer chains and improved dispersion of GO was achieved. They observed that the stiffness and thermal stability of the nanocomposite were improved but the elongation-at-break was decreased by the incorporation of GO in WPU.

Other studies investigated the effect of the amine group containing GO (amineGO) [7,16]. In these investigations, a chemical reaction between -NH_2_ on amine-GO and NCO in WPU enhanced the dispersion of GO in WPU. Hu et al. prepared four different amine (ethylenediamine, triethylenetetramine, 3-aminopropyltriethoxysilane (APTES), and 3-aminopropyl trimethoxy-silane) functionalized GOs and fabricated WPU/amineGO nanocomposites by in situ polymerization [16]. Hydrophobicity, thermal stability, and stiffness of the nanocomposites were improved by the incorporation of amineGO but elongation-at-break was slightly reduced. Lei et al. modified GO with APTES [7]. The silane in APTES and OH in the GO were connected by a hydrolysis reaction, and the modified GO and WPU were connected by the reaction between -NH_2_ in APTES and -NCO in WPU. The GO chemically attached in the WPU was then reduced by diethanol amine and rGO-WPU was obtained. They did not report the electrical conductivity. Furthermore, the physical properties of the composite were not superior to those of the APTES modified GO/WPU investigated by Hu et al.

Since conventional PU is not soluble or easily microdispersed in an aqueous medium, a high content of hydrophilic groups should be copolymerized with the conventional monomers used in the PU. Dimethylolpropanic acid (DMPA) neutralized by trimethylamine (TEA) is frequently used for that purpose, and serves as an internal emulsifier. Most WPUs in the literature were prepared from di-isocyanates, diols, dimethylol propionic acid (DMPA), and triethylamine (TEA). The WPUs thus prepared are anionic (-COO^−^) and GO is anionic as well. Thus, it is expected that WPU and GO are not compatible since a repulsive force acts between the same ionic charges in WPU and GO. Without modification of the GO, a stable GO/anionic WPU dispersion cannot be obtained. Initially dispersed GO in WPU tends to reaggregate in several days [17].

In this study, we adopted a cationic surfactant to improve the affinity between anionic GO and anionic WPU. We also fabricated cationic WPU, and the cationic WPU/anionic GO nanocomposite was prepared. The MWCNT/WPU nanocomposite was also investigated. We prepared oxidized MWCNT (CNTO) in acid, CNTO/cation surfactant/anionic WPU, and CNTO/cationic. The stability of the dispersion of GOs and the physical properties of the nanocomposites were investigated.

## 2. Materials and Methods

### 2.1. Materials

Natural graphite of an average particle size of 6 μm was obtained from Graphit Kropfmühl GmbH (grade name: Cond 5, Hauzenberg, Germany). MWCNT used was Nanocyl NC7000^TM^ (carbon purity >90%, diameter: 9.5 nm, and length: 1.5 μm, Nanocyl, Sambreville, Belgium,). It was used without further treatment. Poly(tetramethylene glycol) (PTMG1000, 1000 g/mol, Aldrich Chemicals, Seoul, Korea) and methyl diethanolamine (MDEA, Aldrich Chemicals, Korea) were dried and degassed at 60 °C under a vacuum for 12 h. Triethylamine (TEA, Junsei Chemicals, Tokyo, Japan) was dried on a 4 A° molecular sieve before use. Dimethylol propionic acid (DMPA; Aldrich Chemicals, Seoul, Korea) was dried at 50 °C for 2 days in a vacuum oven. Isophorone diisocyanate (IPDI; TCI, Tokyo, Japan), dimethyl acetamide (DMAc, Aldrich), 1,4-butanediol (BD, Kanto Chemicals, Tokyo, Japan), dibutyltin dilaurate (DBTDL, Aldrich), benzyl dimethyl hexadecyl ammonium chloride (BDHd-AC, Aldrich, abbreviated as Φ-N^+^(Me)_2_(Et)_15_MeCl^−^, where Φ-, Me, and Et are phenyl, -CH_3_, and -CH_2_CH_3_, respectively), HCl (35–37%, Samchun Pure Chemical Co., Seoul, Korea), concentrated H_2_SO_4_ (98%, Junsei), acetone (Aldrich), KMnO_4_ (99.3%, Samchun), and H_2_O_4_ (35%, Samchun) were used as received. DMPA solution (in DMAc, 20 wt %) and MDEA solution (in acetone, 20 wt %) were prepared before use.

### 2.2. Preparation of GO, CNTO, Cationic Surfactant Treated GO (Cation-GO), and Cationic Surfactant Treated CNTO (Cation-CNTO)

GO nanosheets were fabricated by the oxidation of graphite via the Hummer’s method [18]. Oxidation of the MWCNTs was carried out using sulfuric acid and an oxidizing agent. Of the MWCNTs 1.0 g was mixed with 330 mL of sulfuric acid (98%) by mechanically stirring in a 500 mL flask for 20 min at 90 rpm and for another 15 min at 150 rpm. The temperature was maintained at 0 °C in an ice bath. Of KMnO_4_ 6 g was then added gradually for 30 min and stirred for another 12 h. Finally, 46 mL of deionized water was added to stop the oxidation process. Later, an H_2_O_2_ solution was added and stored for 2–3 days. The supernatant was removed and the product after oxidation was centrifuged and washed with ethanol until the pH stabilized at 6.5. The product was then dried in a vacuum oven to obtain oxidized MWCNT (CNTO).

Subsequently, 1 g of GO (or CNTO) was dispersed in 500 mL of water by ultrasonication for 3 h at 25 °C. Cationic surfactant BDHd-AC (0.5 g) was added into the GO (or CNTO) suspension and stirred for 1 h. Dichloromethane, CH_2_Cl_2_ (200 mL) was added and stirred for another one hour. After the mixture was transferred to a separatory funnel and stored for a few minutes, the supernatant liquid was removed. The precipitate was purified by repeating the following cycle (five times): the addition of deionized water, shaking for dispersion, centrifugation, and removal of the supernatant liquid. The CH_2_Cl_2_ was then removed from the precipitate by washing with CH_3_OH several times in a separatory funnel. The product was dried in a vacuum oven for 24 h to obtain the cationic surfactant treated GO (cation-GO).

### 2.3. Preparation of the WPU, WPU/GO, and WPU/CNTO Nanocomposite 

The reaction was carried out in a four necked, round-bottom flask equipped with a mechanical stirrer, condenser, and dropping funnel under a dry nitrogen atmosphere in a constant temperature oil bath at 75 °C. The reaction scheme is shown in Scheme 1. We prepared six different WPUs that had various compositions of polyol (PTMG), isocyanate (IPDI), and ionomeric monomer (DMPA for anionic and MDEA for cationic). The compositions are listed in Table 1. In all the compositions, the amounts of IPDI and BD were fixed while those of PTMG and ionomeric monomers were varied. Therefore, we made a sample code for the compositions based on the relative amounts of PTMG and ionomeric monomers. In the sample code WPU-P10a10, the P10 and a10 indicate that a PTMG of 10 and anionic monomer of 10 were used. In same manner, cWPU-P9c11 indicates that a PTMG of 9 and cationic monomer (MDEA) of 11 were used.

To prepare the NCO-terminated prepolymer, a predetermined amount of PTMG and IPDI was transferred to the reactor and stirred for a few minutes to equilibrate the temperature in the reactor. The catalyst DBTDL (0.1 g) was added to start the reaction. The reaction proceeded for about one hour until the theoretical NCO content (listed in Table 1 for each compositions) was reached (NCO-terminated PU prepolymer). The change in the NCO content during the reaction was measured using the standard dibutylamine back-titration method (ASTM D1638).

When the theoretical NCO content was reached, the ionomeric monomer (DMPA in DMAc (1/4.85 (g/g)) for anionic or MDEA in acetone (1/6.30 (g/g)) for cationic) was added to the PU prepolymer. The reaction proceeded for 2–3 h until the -OH FTIR peaks in the PTMG completely disappeared. A neutralizer (TEA in water (1/19.80 (g/g)) for anionic or HCl in water (1/58.30 (g/g)) for cationic) was then fed dropwise into the reactor (ionic PU prepolymer). Subsequently, a BD solution in water (1/39.60 (g/g)) was charged dropwise to the reactor for the final chain extension reaction until the -NCO FTIR peaks completely disappeared. The solid content of WPU thus prepared was about 26–28 wt %. WPU prepared from DMPA was anionic WPU (aWPU) since it contains -COO^−^ in the PU main chain while WPU from MDEA contains -N(CH_3_)H^+^- in the main chain and is cationic WPU (cWPU).

For the preparation of WPU/GO and WPU/CNTO nanocomposites, a predetermined amount of GO (or CNTO) was added to the WPU dispersion. The mixture was sonicated for 0.5 h and stirred for 1 h. The mixture was transferred to a glass plate and dried at 25 °C for 24 h and at 60 °C for another 24 h, and finally vacuum-dried at 60 °C for one day to obtain the composite films.

### 2.4. Characterization

Fourier transform infrared (FTIR) analysis was performed on a PerkinElmer 1000 FTIR spectrometer (PerkinElmer, Waltham, MA USA) at room temperature with a resolution of 1 cm^−1^ in a transmission mode. The number of the scan was 10. The mechanical test of the nanocomposites was carried out in a universal testing machine (Hounsfield H10KS, Redhill, UK) with a cross head speed of 100 mm/min. Thin films (60 mm × 10 mm × 0.2 mm) were used for mechanical tests. A dynamic mechanical analysis (DMA) of the WPUs and the nanocomposites was carried out in a Q800 DMA analyzer (TA Instruments, New Castle, DE, USA) at a heating rate of 2 °C from −50 to 150 °C. The test mode was a single cantilever with a frequency of 1 Hz and amplitude of 5%.

## 3. Results and Discussion

As mentioned previously, WPU is a colloidal dispersion of polyurethane in an aqueous medium. The typical solid PU content was 30 wt %, and the sizes of the dispersed PU particle ranged from several tens of nanometers to one hundred nanometers. After drying the dispersion solution, a thin film of PU was formed for coating and binder applications. The WPU/GO nanocomposite was a mixture of dispersed GO and PU particles in water. The stability of the WPU/GO dispersions was thus very important in the storage step before applying in coating and binder applications. Figure 1 exhibits photos of the WPU/GO and WPU/CNTO dispersions from several hours to one-year storage.

The aWPU/CNTO and aWPU/GO dispersions showed poor stability compared to the other four dispersions. CNTO aggregated and sank to the bottom of the vial within several days (Figure 1d). aWPU/GO showed slightly better stability than aWPU/anionic CNTO (aCNTO) but still sank within several days (Figure 1a). In contrast, aWPU/cation-GO (Figure 1b) and aWPU/cation-CNTO (Figure 1e) showed much better stability. They showed homogeneous and stable dispersions even after one year. In the case of aWPU/cation-CNTO, a small amount of sediment was found on the bottom of the vial after one year. However, most cation-CNTOs were floating in the aqueous medium. 

It is certain that the COO^−^ in the GO and R(Me)_2_N^+^-Φ (from cationic surfactant) were connected by an ionic bond as depicted in Figure 2a. Ionic bonds were also formed between the COO^−^ (from DMPA) and (Et)_3_N^+^ (from TEA) during the preparation of WPU (Figure 2b). In this situation, it is most likely that ionic groups gather and build ionic clusters (like a cluster in an ionomer) as depicted in Figure 2d. Strong bonds can be then established between the cation-GO (or cation-CNTO) and aWPU, and stability is enhanced. Ionic clusters similar to those described above were reported in other studies [19,20,21,22].

cWPU/GO showed stable dispersion up to three months after preparation (Figure 1c). After that, the GO particles gradually sank and showed a completely phase separated structure after one year. In the cWPU, MDEA and HCl were used instead of the DMPA and TEA used for the aWPU. Thus, cWPU was cationic (containing N^+^) in the main chain as shown in Figure 2c, and the COO^−^ of the GO had strong ionic bonds with the cationic cWPU as depicted in Figure 2e.

It is believed that cWPU/GO does not form ionic clusters since the GO does not contain ionic groups. The O-H in GO has a covalent, not ionic bond (×). Due to the absence of ionic clusters, the cWPU/GO (and cWPU/CNTO) showed lower dispersion stability than aWPU/cation-GO (or aWPU/cation-CNTO). In addition, we observed that the GOs deteriorated after storage for several months. Bulk density (lightness of GO flake) and color were significantly changed.

We subsequently performed an FTIR analysis of the GO. Figure 3 shows the FTIR spectra. The intensity of the functional peaks (-COOH, -OH, and =O) that appeared in the GO was significantly reduced after three months. This implies that the functional groups had deteriorated during the storage step. It was reported that GO was photoreactive. GO can react under simulated sunlight exposure, forming fragmented photoproducts [23,24]. The reduced number of functional groups is responsible for the reduced stability of the cWPU/GO after about three months. In contrast, the cation-GO did not show such deterioration of the functional groups. It is clear that the cationic surfactant improved the stability of the aWPU/cation-GO via the improved chemical stability of the functional group.

In aWPU/GO (or aWPU/CNTO) nanocomposites, aWPU contains -COO^−^ N^+^H^−^Me(EtOH)_2_ and GO (or CNTO) has carboxylic acid (-COO^δ−^H^δ+^). Since both groups are anionic and have no interaction with each other, they show poor dispersion stability. In contrast, the aWPU/cation-GO and cWPU/GO have increased interaction due to the opposite ionicity of the matrix and fillers. We will refer to this method as ion association in a later section.

DMA (dynamic mechanical analysis) provides insight into the influence of the nanofiller on the molecular mobility of polymers and thus the interactions between a filler and polymer. The tan δ of all the WPU/nano carbon (GO and CNTO) composites vs. temperature at various filler contents were obtained. In all nanocomposites investigated, two transitions were observed: a distinct transition peak at higher temperature and a very weak shoulder at −40 °C. These temperatures were assigned to the glass transition temperature of the soft segment (*T*_gs_) and hard segment (*T*_gh_) [17]. *T*_gh_ varied over a wide range and appeared at 0–60 °C depending on the composition and ionicity of the WPU, while *T*_gs_ was independent of the composition and the iconicity and appeared at −40 °C.

In our previous study [17], we observed that the *T*_gs_ of WPU was not substantially affected by the addition of GO. On the contrary, a distinct change in *T*_gh_ with the addition of the NF was observed. In WPU, soft segments consist of -(CH_2_CH_2_CH_2_CH_2_-O)_n_- coming from the PTMG, and hard segments consist of ionomeric monomers and IPDI connected by the urethane linkages. Therefore, hard segments are highly hydrophilic, while soft segments are much less hydrophilic. GO, which contains several oxygen based functional groups such as carboxylic acid, carbonyl, etc., is likely to be located in the hydrophilic hard segments rather than in the soft segments.

The *T*_gh_ of the nanocomposites thus defined by the DMA analysis are presented in Figure 4 (DMA curves are not shown for space consideration). The *T*_gh_ of pure WPU was influenced by the compositions of the polyol and ionomeric segment. The WPU with higher amounts of polyol provided a lower glass transition temperature since polyol composes the soft segment in the polyurethane [25]. It was also observed that the *T*_gh_ of WPU/CNTO was higher than those of WPU/GO at low concentration of CNTO while the mechanical properties of WPU/CNTO were lower than those of WPU/GO as shown in Table 2 and Figure 4. The higher *T*_gh_ of WPU/CNTO could be attributed to the larger surface area of CNTO compared to that of GO. However, due to entanglement of the CNTO, the mechanical properties of the WPU/CNTO were lower than those of the WPU/GO, as will be discussed in a later section.

It is clearly seen that *T*_gh_ increased with the amount of nanofillers (NF), reached a maximum, and then decreased slightly with a further increase of the NF. The concentration of NF at the maximum *T*_gh_ was influenced by the concentration of the ionomeric segments (-COO^−^ N^+^H^−^Me(EtOH)_2_) from DMPA/TEA for aWPU and -N^+^H (Me)^−^ from MDEA/HCl for cWPU). The higher the content of the ionomeric segment is, the higher the NF content at maximum *T*_gh_. In this work, we produced a homogeneous and stable dispersion of the WPU/NF composite using the anion-cation association (anionic WPU/cationic NF or cationic WPU/anionic NF). The ionicity of the WPU was produced by incorporating an ionomeric segment. Thus, as the concentration of ionomeric segments in WPU increases, the maximum NF content, which can be associated to WPU, increases.

The nanostructure of fillers strongly affects the mechanical properties of nanocomposites. Enhanced mechanical properties are obtained when the nanofiller dispersion is stable and homogeneous. From the stability test and DMA analysis, it was evident that the cation-anion association was a good way to achieve a homogeneous and stable dispersion of NFs in WPU.

Figure 5 shows stress-strain curves of the WPU/NF nanocomposites with various NFs. Among all NFs, cation-GO showed the highest mechanical properties in anionic WPU, and anionic GO showed the highest mechanical properties in cationic WPU. aWPU/cation-CNTO and cWPU/anionic CNTO also showed relatively good mechanical properties but their mechanical properties were slightly lower than those of aWPU/cation-GO and cWPU/GO.

Due to the high aspect ratio of CNTO and its entangled nature, dispersion of the CNTs was worse than that of GO, and thus, their mechanical properties were lower than the GO based composites. It is evident that the ionically associated nanocomposites showed higher mechanical properties. The aWPU/GO nanocomposites that contained identically charged ions between the matrix and filler showed even lower mechanical properties than those of pristine WPU.

The tensile test results for aWPU/cation-GO, aWPU/cation-CNTO, cWPU/GO, and cWPU/CNTO are summarized in Table 2. It was observed that the initial modulus of the composites continuously increased with the NF content but the elongation at break (EB) and tensile strength (TS) at break increased with the NF content, reached the maximum value, and then decreased slightly with a further increase of the filler content. This trend is the same as that of *T*_gh_ vs. the NF content. Maximum EB and maximum TS were observed at the same NF content where the maximum *T*_gh_ was observed. Beyond the NF content of the maximum *T*_gh_, the remaining NFs not participating in the ion association did not contribute to the increase in mechanical properties, but appeared to reduce the mechanical properties and *T*_gh_. The remaining NFs most likely aggregated by themselves. Thus, it would be more natural that aggregated (or phase separated) NF does not affect the glass transition of WPU. These unusual results may be due to polymer molecular scission by the remaining GOs, which will be investigated in future work.

According to the discussion on ion association, the ratio of NF content at the maximum *T*_gh_ and the concentration of ionomeric segment should be constant regardless of the composition of WPU (ionomeric segment content in WPU). For example, the NF contents at maximum *T*_gh_ were 1, 0.5, and 0.3 wt % for aWPU-P10a10, aWPU-P11a9, and aWPU-P12a8, respectively. The relative amount of ionomeric segments in aWPU-P10a10, aWPU-P11a9, and aWPU-P12a8 were 1, 0.9, and 0.8 (the notation WPU-a10P10 implies a relative loading of 10 mol of polyol and 10 mol of DMPA). Since -COO^−^ N^+^H(Me)_2_-R in cation-GO and -COO^−^ N^+^H(Et)_3_ in aWPU associate, it is expected that the GO content at maximum *T*_g_ was also 1, 0.9, and 0.8 for aWPU-P10a10, aWPU-P11a9, and aWPU-P12a8. The discrepancy between 1:0.5:0.3 and 1:0.9:0.8 was not clear at this point. Experimental results implied that more GOs associated with one ionomeric segment as the content of the ionomeric segment increased. Knowledge of the detailed association structure will help to illuminate this issue.

Many studies in the literature have investigated anionic WPU/GO (or rGO) nanocomposites. Their mechanical properties were low because of the poor dispersion that originates from the same ionic charges of GO (or rGO) and aWPU. In this study, we overcame those problems with cationic WPU, which has an ionic charge opposite to that of GO, and a cationic surfactant to increase the compatibility between WPU and GO for the first time. Between the two nanocomposites, aWPU/cation-GO was considered a better nanocomposite since GO had lower storage stability than cation-GO. cWPU/GO had fairly good mechanical properties just after preparation, but its mechanical properties were expected to be worse after a long period of storage due to the reduced chemical stability of the GO itself.

## 4. Conclusions

GO (which is anionic due to the -COOH) and aWPU were observed to be not compatible because of their identical ionic charges. Consequently, aWPU/GO nanocomposites showed poor dispersion stability and mechanical properties. These poor dispersion stability and mechanical properties were improved to a great extent when the cationic surfactant modified GO (cation-GO) was incorporated in the aWPU. This improvement was found to be due to the higher affinity of the cation-GO with aWPU than GO. It was also proved that the high affinity between the cation-GO and aWPU was caused by the opposite ionic charges in cation-GO and aWPU. cWPU/GO, cWPU/CNTO, and aWPU/cation-CNTO also showed good dispersion stability and mechanical properties. When the NF and WPU had opposite ionic charges, the nanocomposites showed higher dispersion stability and consequently high mechanical properties.

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
