# Peer review of "Effect of a Monomer Composition on the Mechanical Properties and Glass Transition Temperature of a Waterborne Polyurethane/Graphene Oxide and Waterborne Polyurethane/MWCNT Nanocomposite"

_polymers, 2020, doi:10.3390/polym12092013_

Round 1

Reviewer 1 Report

WPU’s are basically two phase colloidal systems where PU’s are dispersed in an aqueous medium. WPUs are widely used in coatings and adhesives because they are environmentally friendly. A rapid scifinder search of Anionic waterborne polyurethane shows a large number of articles published, with an increasing trend in recent years (i.e. 10K were published in 2019, and 4145 have been published during 2020). These numbers are a clear expression of the thematic importance in the materials field.

However, please find enclosed some of my recommendations and comments, after revised the present work

  1. Authors must be careful with self-plagiarism. Urkund software showed an 18% concentrated mainly in the following sources

https://www.jim.or.jp/journal/e/58/06/892.html

https://www.jstage.jst.go.jp/article/matertrans/58/6/58_M2016411/_html/-char/ja

https://www.jstage.jst.go.jp/article/matertrans/58/6/58_M2016411/_html/-char/en

https://www.researchgate.net/publication/229925271_Preparation_and_properties_of_acid-treated_multiwalled_carbon_nanotubewaterborne_polyurethane_nanocomposites

  1. Another problem is the writing. There are many typos, grammatical errors, and editing issues, as illustrated below. It is still necessary for the authors to make every endeavor to improve the manuscript preparation before it is accepted for publication.

Generals

Consider inserting a comma to separate the elements. (i.e. Line 27, 29, 37, 48, 50, 59,

The word nanocomposites is used also as nano-composite through all the text, please unified.

The abbreviation et. al. seems to be incorrectly punctuated. Consider changing the punctuation to “et al. ”

The authors use the phrase “very poor”. When this kind of phrase are used, it is necessary to define parameters or standards. Is it very poor in comparison to what?

Line 12 Replace “…because of repulsive force…” by “…because of the repulsive force…”

Line 13 Added a comma after GO.

Line 17 Replace “…showed maximum …” by “…showed the maximum …”

Line 35 Replace “…by incorporation  …” by “… by the incorporation …”

Line 46 Replace “…dispersion  …” by “… the dispersion …”

Line 64 Include an “a” before “chemical”

Line 70, 73, 86, 172, 201, 258, 287 302, 303 inserting a comma before “and. ”

Line 80 Replace “di-isocianates” by “di-isocyanates”

Line 241  Replace “…consist  …” by “… consists …”

Line 242 Added a comma before while.

Line 243 Replace “…carbonyl and etc, …” by “… carbonyl, etc., …”

Line 250 Replace “…is …” by “… are …”

Line 290 Added “the” before “maximum”

Line 296 Replace “…that …” by “… than …”

Line 296 Replace “…do…” by “… does …”

Line 325 A final sentence dot is missing.

  1. In the last paragraph of Introduction, the authors stablished

“For comparison, we prepared oxidized MWCNT (CNTO) in acid, CNTO/cation 87 surfactant/anionic WPU and CNTO/cationic WPU were prepared. ”

Comparison of what? I consider it essential to define the parameters or aspects to be compared, or at least which properties are interesting to compare.

  1. Scheme 1 needs to be improved to a better understanding.
  2. I recommend including the labels A, B, C ... used in the figure 1, in the paragraph of Lines 184-190. It was hard to read and understand this part
  3. Figure 2a needs to be improved. Please, revise the ionic bond represented. In my opinion, the ionic bond between COO- and the quaternary amine is not correctly represented. The negative charge is delocalized between the two oxygens of carboxylate anion, but charge would never be on central Carbon.
  4. Line 208 The authors established

“The O-H in GO has a covalent, not ionic bond. ” In my opinion, this sentence is not necessary because all O-H bonds are covalents, despite their more or less ionic character.

  1. The FTIR analysis (Lines 212-218) showed a direct relation between the stability of suspensions through the time and the deterioration of functional groups present in GO. I recommend adding a brief explanation related to the reasons for the degradation suffered by functional groups
  2. Line 225, please revise the structures, the charges should be place as superscript.
  3. Line 234 “transition peak at higher temperature (0 ~ 60 oC) ” What does 0 ~ 60 oC mean?
  4. Lines 258-266 Why Tgh values reach a maximum and decrease slightly with further increase of the NF? Besides, the last sentence of this paragraph is not clear.
  5. The conclusions need to be exhaustively revised and rewritten. It is not clear how these conclusion answer the main goal defined to this research.

Author Response

WPU’s are basically two phase colloidal systems where PU’s are dispersed in an aqueous medium. WPUs are widely used in coatings and adhesives because they are environmentally friendly. A rapid scifinder search of Anionic waterborne polyurethane shows a large number of articles published, with an increasing trend in recent years (i.e. 10K were published in 2019, and 4145 have been published during 2020). These numbers are a clear expression of the thematic importance in the materials field.
However, please find enclosed some of my recommendations and comments, after revised the present work

1. Authors must be careful with self-plagiarism. Urkund software showed an 18% concentrated mainly in the following sources
https://www.jim.or.jp/journal/e/58/06/892.html
https://www.jstage.jst.go.jp/article/matertrans/58/6/58_M2016411/_html/-char/ja
https://www.jstage.jst.go.jp/article/matertrans/58/6/58_M2016411/_html/-char/en
https://www.researchgate.net/publication/229925271_Preparation_and_properties_of_acid-treated_multiwalled_carbon_nanotubewaterborne_polyurethane_nanocomposites

Answer: First three links above are based on the same paper, which is our previous work. We checked the self-plagiarism with CopyKiller and modified some of the paragraphs in the paper. It shows 9% of plagiarism rate. Most paragraphs pointed by the ‘CopyKiller’ was in the Experimental Section. Our previous work and this work used similar materials and apparatus. We think that the 9 % in CopyKiller is acceptable rage.

2. Another problem is the writing. There are many typos, grammatical errors, and editing issues, as illustrated below. It is still necessary for the authors to make every endeavor to improve the manuscript preparation before it is accepted for publication.

Generals
Consider inserting a comma to separate the elements. (i.e. Line 27, 29, 37, 48, 50, 59,
The word nanocomposites is used also as nano-composite through all the text, please unified.
The abbreviation et. al. seems to be incorrectly punctuated. Consider changing the punctuation to “et al. ”
The authors use the phrase “very poor”. When this kind of phrase are used, it is necessary to define parameters or standards. Is it very poor in comparison to what?

Line 12 Replace “…because of repulsive force…” by “…because of the repulsive force…”
Line 13 Added a comma after GO.
Line 17 Replace “…showed maximum …” by “…showed the maximum …”
Line 35 Replace “…by incorporation  …” by “… by the incorporation …”
Line 46 Replace “…dispersion  …” by “… the dispersion …”
Line 64 Include an “a” before “chemical”
Line 70, 73, 86, 172, 201, 258, 287 302, 303 inserting a comma before “and. ”
Line 80 Replace “di-isocianates” by “di-isocyanates”
Line 241  Replace “…consist  …” by “… consists …”
Line 242 Added a comma before while.
Line 243 Replace “…carbonyl and etc, …” by “… carbonyl, etc., …”
Line 250 Replace “…is …” by “… are …”
Line 290 Added “the” before “maximum”
Line 296 Replace “…that …” by “… than …”
Line 296 Replace “…do…” by “… does …”
Line 325 A final sentence dot is missing.

Answer: Corrections were made according to reviewer’s comments. English was reviewed by a native speaker.

3. In the last paragraph of Introduction, the authors stablished
“For comparison, we prepared oxidized MWCNT (CNTO) in acid, CNTO/cation 87 surfactant/anionic WPU and CNTO/cationic WPU were prepared. ”
Comparison of what? I consider it essential to define the parameters or aspects to be compared, or at least which properties are interesting to compare.

Answer: The paragraph was modified to clarify the meaning of the sentence.

4. Scheme 1 needs to be improved to a better understanding.

Answer: Scheme 1 was be improved.

5. I recommend including the labels A, B, C ... used in the figure 1, in the paragraph of Lines 184-190. It was hard to read and understand this part
Answer: The paragraph was modified according to reviewer’s comment.

6. Figure 2a needs to be improved. Please, revise the ionic bond represented. In my opinion, the ionic bond between COO- and the quaternary amine is not correctly represented. The negative charge is delocalized between the two oxygens of carboxylate anion, but charge would never be on central Carbon.

Answer: Figure 2a was modified according to reviewer’s comment.

7. Line 208 The authors established
“The O-H in GO has a covalent, not ionic bond. ” In my opinion, this sentence is not necessary because all O-H bonds are covalents, despite their more or less ionic character.

Answer: The sentence was removed.

8. The FTIR analysis (Lines 212-218) showed a direct relation between the stability of suspensions through the time and the deterioration of functional groups present in GO. I recommend adding a brief explanation related to the reasons for the degradation suffered by functional groups

Answer: Explanation for the degradation of GO was added.

9. Line 225, please revise the structures, the charges should be place as superscript.

Answer: Errors were corrected throughout the text.

10. Line 234 “transition peak at higher temperature (0 ~ 60 oC) ” What does 0 ~ 60 oC mean?

Answer: Expression was modified to clarify the context.

11. Lines 258-266 Why Tgh values reach a maximum and decrease slightly with further increase of the NF? Besides, the last sentence of this paragraph is not clear.

Answer: Tgh increases with NF content up to the critical NF content, and decreases beyond the critical NF content. We think that excess NF does not participate in the anion-cation association but aggregates and acts like a plasticizer. The last sentence were modified to clarify the context.

12. The conclusions need to be exhaustively revised and rewritten. It is not clear how these conclusion answer the main goal defined to this research.

Answer: Conclusion was revised.

Reviewer 2 Report

This manuscript shows the effect of monomer composition on the stability, mechanical properties and glass transition temperature of hybrid waterborne polyurethane (WPU) nanocomposites. The contents are interesting for readers in Polymers. If appropriate revisions are performed, the revised paper may be suitable for publication in Polymers. I have some comments on this manuscript.

- P.1, Line 14: It should be "glass transition temperature" instead of “glass transition”.

It should be " nanocomposites" instead of “nano-composites”.

- P.2, Line 93-94: “molecular weight = 1,000 g/mol”. Needs to be corrected.

- Details of MWCNT are missing in Section 2.1 Materials.

- Why choose carbon nanotubes as comparative experimental materials?

- GO or rGO can endow polyurethane many excellent properties (such as flame retardancy, electrical conductivity, mechanical properties). Why does the article focus on the electrical conductivity (in P.2, Line 48-56) that is not involved in the following experimental tests?

- I have a certain skepticism towards “They did not report the electrical conductivity and this is probably because of the low electrical conductivity of the nanocomposites investigated.” Need to provide proof or delete it.

- Characterization techniques need more information so that the results can be reproduced by someone in the field some specifics below include but are not limited to: FTIR characterization technique needs more information: test temperature, mode, number of scans and the resolution. The mode, sample size, test frequency, vibration amplitude, and test temperature range for DMA testing also need to be supplemented.

- From the data analysis and result description of Figure 1 and Figure 2, the article lacks necessary particle diameters and Zeta potentials tests. In addition, scientific papers cannot make hypothetical descriptions based on macroscopic phenomena, but require detailed comparative analysis from the perspective of microscopic mechanisms. For the analysis of the ionic interaction between nanofillers and WPU, it is recommended that the author refer to literature “Macromolecules 39 (2006) 6133–6141” and “10.1016/j.porgcoat.2019.105273”.

- The FTIR baseline needs to be leveled, otherwise the test should be re-tested. It is necessary to select the peak of a certain group as the reference peak and normalize before the semi-quantitative characterization of FTIR.

- In order to increase the credibility of opinions, it is necessary to supplement directly related references, some of which are listed below:

  1. P.1, Line 34-35: “In spite of……some fields.”
  2. P.2, Line 48-49: “Many studies in……of WPU.”
  3. P.7, Line 236-237: “Tgh varied over……and the ionicity.”

Author Response

This manuscript shows the effect of monomer composition on the stability, mechanical properties and glass transition temperature of hybrid waterborne polyurethane (WPU) nanocomposites. The contents are interesting for readers in Polymers. If appropriate revisions are performed, the revised paper may be suitable for publication in Polymers. I have some comments on this manuscript.

- P.1, Line 14: It should be "glass transition temperature" instead of “glass transition”.
It should be " nanocomposites" instead of “nano-composites”.

 Answer: Error was corrected.

- P.2, Line 93-94: “molecular weight = 1,000 g/mol”. Needs to be corrected.
 Answer: Corrected.

- Details of MWCNT are missing in Section 2.1 Materials.
 Answer: Information on the MWCNT was added.

- Why choose carbon nanotubes as comparative experimental materials?
 Answer: The paragraph was modified to clarify the meaning of the sentence.

- GO or rGO can endow polyurethane many excellent properties (such as flame retardancy, electrical conductivity, mechanical properties). Why does the article focus on the electrical conductivity (in P.2, Line 48-56) that is not involved in the following experimental tests?
 Answer: The paragraph was modified to improve the understandability.

- I have a certain skepticism towards “They did not report the electrical conductivity and this is probably because of the low electrical conductivity of the nanocomposites investigated.” Need to provide proof or delete it.
 Answer: The sentence, “conductivity and this is probably because of the low electrical conductivity of the nanocomposites investigated.”, was deleted.

- Characterization techniques need more information so that the results can be reproduced by someone in the field some specifics below include but are not limited to: FTIR characterization technique needs more information: test temperature, mode, number of scans and the resolution. The mode, sample size, test frequency, vibration amplitude, and test temperature range for DMA testing also need to be supplemented.
 Answer: Experimental details were added

- From the data analysis and result description of Figure 1 and Figure 2, the article lacks necessary particle diameters and Zeta potentials tests. In addition, scientific papers cannot make hypothetical descriptions based on macroscopic phenomena, but require detailed comparative analysis from the perspective of microscopic mechanisms. For the analysis of the ionic interaction between nanofillers and WPU, it is recommended that the author refer to literature “Macromolecules 39 (2006) 6133–6141” and “10.1016/j.porgcoat.2019.105273”.
 Answer: Thank you for comments and the reference was added.

- The FTIR baseline needs to be leveled, otherwise the test should be re-tested. It is necessary to select the peak of a certain group as the reference peak and normalize before the semi-quantitative characterization of FTIR.
Answer: Not like other organic compounds, GO does not have a reference peak which is not changed with progress of reaction or degree of functionality. GO is known to be photo reactive [M. Shams, L. M. Guiney, L. Huang, M. Ramesh, X. Yang, M. C. Hersam and I. Chowdhury ,Influence of functional groups on the degradation of graphene oxide nanomaterials , Environ. Sci.: Nano, 2019, 6, 2203–2214]. At the moment of GO synthesis, it is yellow. However it became dark brown after 2 ~ 3 days depending on the storage condition. Our GO samples taken for the FTIR was already deteriorated. Thus, we cannot help but using the current spectra.  

- In order to increase the credibility of opinions, it is necessary to supplement directly related references, some of which are listed below:

1. P.1, Line 34-35: “In spite of……some fields.”

Answer: Reference was added.

2. P.2, Line 48-49: “Many studies in……of WPU.”

Answer: References were added.

3. P.7, Line 236-237: “Tgh varied over……and the ionicity.”

Answer: This sentence describes our experimental result on Fig. 4. It is not a general knowledge.

Reviewer 3 Report

This work introduces an interesting quasi in situ polymerization of WPU composites. The design and process of this work are impressive and well-finished. The results also confirm that this route works for improving Tg and mechanical property of WPU composites. I suggest this work could be accepted after addressing some trivial questions.

  • First, the writing of this manuscript should be improved. If possible, it would be better if revised by a native English speaker.
  • The style of the figures should be identical, for example, In Figure 1, using A, B, C..., others use (a), (b),... Others including colour and formats also need to be improved.

Author Response

This work introduces an interesting quasi in situ polymerization of WPU composites. The design and process of this work are impressive and well-finished. The results also confirm that this route works for improving Tg and mechanical property of WPU composites. I suggest this work could be accepted after addressing some trivial questions.

- First, the writing of this manuscript should be improved. If possible, it would be better if revised by a native English speaker.
Answer: Manuscript was reviewed by a native speaker.

- The style of the figures should be identical, for example, In Figure 1, using A, B, C..., others use (a), (b),... Others including colour and formats also need to be improved.
Answer: Figures was revised according to reviewer’s comment.

Round 2

Reviewer 2 Report

I have read the revised manuscript and its response letter. Based on the reviewers’ questions, the authors revised the manuscript to improve the quality. However, some questions still remain in the current form as follows.

- In the Q1-8, I recommended the authors to add the particle diameters and Zeta potentials tests to scientifically explain the phenomenon in Figure 1 and the possible mechanism in Figure 2. Moreover, I strongly recommended that the author learn the analysis and explanation of the influence of fillers on the particle size and stability of WPU in the “10.1016/j.porgcoat.2019.105273” and “Macromolecules 39 (2006) 6133–6141”. However, the author only cited the literature “Macromolecules 39 (2006) 6133–6141” and did not make substantial changes to this revised manuscript.

Author Response

I have read the revised manuscript and its response letter. Based on the reviewers’ questions, the authors revised the manuscript to improve the quality. However, some questions still remain in the current form as follows.

- In the Q1-8, I recommended the authors to add the particle diameters and Zeta potentials tests to scientifically explain the phenomenon in Figure 1 and the possible mechanism in Figure 2. Moreover, I strongly recommended that the author learn the analysis and explanation of the influence of fillers on the particle size and stability of WPU in the “10.1016/j.porgcoat.2019.105273” and “Macromolecules 39 (2006) 6133–6141”. However, the author only cited the literature “Macromolecules 39 (2006) 6133–6141” and did not make substantial changes to this revised manuscript.

Answer:

In the preparation of WPU/nanofiller composites, we first synthesized WPU. After the polymerization reaction for the WPU was competed by adding butanediol/water solution dropwise to the reactant until the –NCO FTIR peaks completely disappeared, nanofiller was added to the WPU dispersion and the mixture was sonicated for 0.5 h and stirred 156 for 1 h. Therefore, particle size of WPU is already fixed and are not influenced by nanofiller subsequently added.

For example, in Fig. 1 we have six nanocomposites. (a) aWPU-P10a10/GO (1.5 wt. %), (b) aWPU-P10a10/cation-GO (1.5 wt. %) (c) cWPU-P10c10/GO (1.5 wt. %), (d) aWPU-P10a10/CNTO (1.0 wt. %), (e) aWPU-181 P10a10/cation-CNTO (1.0 wt. %) and (f) cWPU-P10c10/CNTO (1.0 wt. %).  

Here, four nanocomposites, (a) aWPU-P10a10/GO (1.5 wt. %), (b) aWPU-P10a10/cation-GO (1.5 wt. %), (d) aWPU-P10a10/CNTO (1.0 wt. %), and (e) aWPU-P10a10/cation-CNTO (1.0 wt. %), contain the same aWPU, meaning that the polyurethane (PU) particle size distributions are identical for these four nanocomposites. These four nanocomposites with same aWPU and identical particle size distribution showed very different dispersion stability.

In same manner, two nanocomposites, (c) cWPU-P10c10/GO (1.5 wt. %) and (f) cWPU-P10c10/CNTO (1.0 wt. %), contain same WPU and have identical PU particle size distribution. These two nanocomposites with same cWPU and identical particle size showed very different dispersion stability.

Thus, we declare that ionicities of WPU and nanofillers plays a major role in dispersion stability.

One more thing worth mentioning is that PU particles in WPU dispersion are stable. PU particles do not aggregate or settle down in the WPU dispersion. It is nanofiller (NF) that determines the stability of WPU/NF nanocomposites, not PU particles. NF could aggregate or settle down depending on the conditions.   

Both aWPU-P10a10/cation-GO (1.5 wt. %) and cWPU-P10c10/GO (1.5 wt. %) have opposite ionicity between NF and WPU and showed different dispersion stability though both showed very high dispersion stability. In this case, we agree that particle size and Zeta potentials play a role to some extent in the dispersion stability. However, we are sure that overall, the ionicity plays more important role than the particle size and Zeta potential.      

Moreover, unfortunately, we cannot perform the particle size and Zeta potentials tests. Experiments for this work were performed mainly in 2016~2017, and the project was terminated 2 years ago. WPU samples were already consumed and some remaining samples (if any) have already been dried and solidified. For the particle size and Zeta potentials test, new WPU samples must be synthesized again. New graduate student needs to be trained and practice the WPU synthesis for this job. Probably, it will take at least six months. Considering that the particle size and Zeta potentials plays minor role in the dispersion stability, performing the test is not cost-effective.